# *C. elegans* Models to Study the Propagation of Prions and Prion-Like Proteins

**DOI:** 10.3390/biom10081188

**Published:** 2020-08-15

**Authors:** Carl Alexander Sandhof, Simon Oliver Hoppe, Jessica Tittelmeier, Carmen Nussbaum-Krammer

**Affiliations:** Center for Molecular Biology of Heidelberg University (ZMBH) and German Cancer Research Center (DKFZ), DKFZ-ZMBH Alliance, Im Neuenheimer Feld 282, D-69120 Heidelberg, Germany; a.sandhof@zmbh.uni-heidelberg.de (C.A.S.); o.hoppe@zmbh.uni-heidelberg.de (S.O.H.); j.tittelmeier@zmbh.uni-heidelberg.de (J.T.)

**Keywords:** prions, prion-like propagation, neurodegenerative diseases, *C. elegans*, intercellular spreading

## Abstract

A hallmark common to many age-related neurodegenerative diseases, such as Alzheimer’s disease (AD), Parkinson’s disease (PD), and amyotrophic lateral sclerosis (ALS), is that patients develop proteinaceous deposits in their central nervous system (CNS). The progressive spreading of these inclusions from initially affected sites to interconnected brain areas is reminiscent of the behavior of bona fide prions in transmissible spongiform encephalopathies (TSEs), hence the term prion-like proteins has been coined. Despite intensive research, the exact mechanisms that facilitate the spreading of protein aggregation between cells, and the associated loss of neurons, remain poorly understood. As population demographics in many countries continue to shift to higher life expectancy, the incidence of neurodegenerative diseases is also rising. This represents a major challenge for healthcare systems and patients’ families, since patients require extensive support over several years and there is still no therapy to cure or stop these diseases. The model organism *Caenorhabditis elegans* offers unique opportunities to accelerate research and drug development due to its genetic amenability, its transparency, and the high degree of conservation of molecular pathways. Here, we will review how recent studies that utilize this soil dwelling nematode have proceeded to investigate the propagation and intercellular transmission of prions and prion-like proteins and discuss their relevance by comparing their findings to observations in other model systems and patients.

## 1. Introduction

### 1.1. The Prion-Like Behavior of Proteins Associated with Human Neurodegenerative Diseases

Proteinopathies are a family of diseases that encompass all pathologies linked to protein misfolding. While this is not an exclusive feature, most of them manifest tissue specific or systemic inclusions composed of one or several disease specific proteins. Among proteinopathies, only transmissible spongiform encephalopathies (TSEs) are infectious and can be transmitted between individuals. These rapidly progressing neurodegenerative diseases occur in sheep (scrapie), cattle (bovine spongiform encephalopathy), deer (chronic wasting disease), felines (feline spongiform encephalopathy), and humans (Creutzfeldt–Jakob disease, fatal familial insomnia, and kuru) [1]. When Stanley B. Prusiner demonstrated that a protein is the causative agent in scrapie, he introduced the term ‘prion’, an acronym for ‘proteinaceous infectious agent’, to distinguish it from other infectious pathogenic vectors containing nucleic acids [2]. Later it was shown that a prion is a ß-sheet rich isoform (PrP^Sc^) of the α-helix rich cellular prion protein (PrP^C^/PRNP) [3,4]. 

The process of prion formation is initiated by the assembly of a small oligomeric nucleus or seed, which can force the native monomer to adopt the same pathological conformation and polymerize into higher order amyloids by incorporating additional monomers at their ends [5,6]. Disease pathology then spreads from an initial site of spontaneous PrP^Sc^ formation or exposure to other areas within the host, which is attributed to the said induced misfolding of native isoforms in neighboring naïve cells through the transmission of prion particles.

More recently, a prion-like propagation mechanism has also been proposed for several other neurodegenerative disease-related proteins that form amyloid deposits in the CNS of affected patients [7]. This assumption is based on observations that such inclusions, which are initially only observed in particular parts of the brain, follow a disease specific spreading pattern affecting neuro-anatomically interconnected brain areas over the course of the disease reminiscent of how TSEs progress over time [8]. In the last decade, evidence for such a prion-like behavior has been shown for several of the most prominent disease proteins, including α-Synuclein/SNCA (α-Syn; associated with Parkinson’s disease (PD) and other synucleinopathies), Tau/MAPT (associated with Alzheimer’s disease (AD), traumatic brain injury (TBI) induced tau pathology, and other tauopathies), the Aβ-peptide (associated with AD), huntingtin with extended polyglutamine stretches (HTT-polyQ; associated with Huntington’s disease (HD)), and TDP-43 (associated with amyotrophic lateral sclerosis (ALS)) [9,10].

Several criteria must be met to qualify as a prion-like protein, such as the ability to form self-propagating aggregates, and transmissibility. Each of the proteins listed above aggregates following a seeded polymerization mechanism [11,12,13,14,15,16]. In vitro studies on the kinetics of the aggregation reaction revealed that the assembly starts with a lag phase during which the initial oligomeric seeds are formed, followed by an exponential log phase of amyloid fiber growth by rapid monomer incorporation. Adding sonicated pre-formed fibers to native monomers seeds their conversion and incorporation into the β-sheet rich amyloid fiber and bypasses the initial lag phase of primary nucleation. This autocatalytic, templated self-assembly, facilitated by the ordered arrangement of the individual units in the amyloid fibril, is the molecular basis for prion-like propagation.

Transmissibility, i.e., dissemination between cells or individuals, has long been considered an exclusive feature of PrP^Sc^. However, the observation that naïve grafted neurons also developed Lewy bodies, intercellular inclusions containing α-Syn [17], in subjects with PD was the first evidence that pathological α-Syn might act similarly and can spread from affected to healthy neurons [18,19]. This led to a spark of experiments in rodent and cell culture models to investigate the cell to cell transmission of amyloidogenic proteins. To date, the induction of inclusion formation by local injection of synthetically generated seeds or extracted patient brain material in transgenic mice, expressing a disease related mutant form of the respective human protein, has been shown for all of the above-mentioned prion-like proteins [20,21,22,23,24].

### 1.2. Caenorhabditis Elegans as a Metazoan Model Organism

Since the spreading of pathology is an important aspect of disease progression, the underlying cellular mechanisms are of increasing scientific interest. However, although it is now widely accepted that α-Syn, Tau, and their fellow culprits are prion-like and able to spread at least between cells within a host, the cellular pathways that mediate the spreading of protein misfolding are poorly understood.

Most initial observations on the progression of neurodegenerative diseases came from histopathological studies with postmortem patient material. This work paved the way for the prion-like hypothesis by identifying the spreading pattern of inclusion body formation [25,26,27]. To validate the seeding and spreading competence of disease proteins and to gain insight into the underlying molecular mechanisms, several cell culture and animal models have been developed. However, standard cell culture models lack the depth and complexity of an intact tissue and using organoids is technically challenging, resulting in considerable variations in their composition. Using mammalian models of neurodegeneration, on the other hand, makes it possible to consider the impact of environmental factors and other cell types and organs as well as connective tissues. Yet, working with mammals is very demanding and expensive, which usually restricts studies to small cohorts. Moreover, studies of prion-like spreading in rodents is in most cases limited to snapshots after sacrificing the animals, as in vivo imaging involves special equipment [28,29].

The soil dwelling non-parasitic nematode *Caenorhabditis elegans* (*C. elegans*) combines several advantages that makes it especially useful to study the prion-like propagation of protein misfolding in a multicellular organism. In a laboratory setting, these animals are kept on agarose plates with a lawn of *E. coli* or other bacteria as food source. Nematode populations are typically grown at temperatures between 15–25 °C and are easily scalable. If a large number of worms is required, cultivation in liquid media can even be carried out in a fermenter [30]. The animal has a short lifecycle and develops through embryogenesis and four larval stages to its reproductive age after merely three days. Adult hermaphrodites produce approximately 300 isogenic offspring in their first third of adulthood and live for about 15–20 days. While hermaphrodites are the predominant sex, males do arise spontaneously (≤0.2%) as a result of meiotic non-disjunction of the X chromosome and can be used for genetic crosses. Adult *C. elegans* are post-mitotic with a set number of exactly 959 somatic cells (hermaphrodite) whose complete developmental lineage from the fertilized oocyte has been traced [31,32,33]. Moreover, the complete neuronal connectome of all neurons (302 in hermaphrodites and 385 in males) has been mapped down to the different cell types and individual synapses [34,35]. Since this small neuronal circuitry controls all of the animal’s complex behaviors (crawling and swimming, chemotaxis, foraging, mating, touch responses and others), changes in its functionality due to genetic perturbations or external factors can be directly investigated by behavioral assays. 

The fully sequenced genome is highly annotated, with ~41% of the protein coding genes having predicted human orthologues [36], including many key genes implicated in neurodegenerative diseases [37]. A toolbox of well-defined promoter sequences and the availability of whole genome RNAi libraries enable precise spatiotemporally controlled alterations of protein expression. This genetics repertoire has been invaluable for the discovery and characterization of conserved cellular pathways that control stress responses, apoptosis, and aging [38,39,40]. Transgenic animals are usually generated by microinjection of the plasmid DNA into the gonadal syncytium, which forms heritable extrachromosomal arrays [41]. Traditionally, these were stably integrated into the organism’s genome by UV or γ-irradiation, leading to the insertion of multiple copies of the transgene at a random site. Nowadays, the adaptation of the CRISPR/Cas system allows for site-specific integrations of a single copy transgene and even more advanced genetic manipulations of the endogenous genome. 

Another notable advantage of this animal is its transparent nature. *C. elegans* can be immobilized and imaged alive using almost any type of microscope, which enables investigating the dynamics of fluorescently tagged transgenic or endogenous proteins in vivo throughout development and aging. Its scalability, genetic malleability, and ease of in vivo microscopy makes *C. elegans* a powerful system and led to the rapid adoption of this nematode to study proteostasis in general and specifically protein aggregation and toxicity associated with human neurodegenerative diseases.

## 2. *C. Elegans* Models to Study Prion-Like Proteins

### 2.1. C. Elegans as a Model System to Study the Toxicity Associated with Disease Proteins

*C. elegans* is considered a well-established model organism for neurodegenerative diseases and is increasingly used to complement studies in mice and cell culture. Nematode models are usually generated by the ectopic expression of wildtype (WT) or mutant forms of human disease-related proteins [42]. By using suitable promoters, transgene expression is restricted to specific tissues, often muscle cells or neurons, which are both post-mitotic and unable to regenerate. Transgenes can even be directed to specific neuron classes, such as dopaminergic neurons or mechanosensory neurons, which allows focusing the investigation on a particular subtype of neurons.

As mentioned above, transgenes are frequently fused to fluorescent tags like green fluorescent protein (GFP) in order to monitor protein dynamics in the living animal [43]. If a direct fluorescent labeling is not desired, the expression of the protein of interest can be tracked indirectly by co-expressing fluorescent proteins under the same promoter. Furthermore, the co-expression of such tissue markers can be used as a non-invasive way to determine the integrity of the respective tissue [44].

Tissue damage in muscles and neurons results in defined phenotypes for which there are several established assays that allow a precise analysis of the toxic effect of aggregates. Movement defects, which can be caused by muscle damage as well as neurodegeneration, can be assessed by motility assays on plates or in liquid medium, whereas impairments in the neuronal perception can be assessed by chemosensory assays [45,46,47].

An important advantage of the worm, which is often exploited in *C. elegans* research, is the ease of unbiased genome-wide screening for genetic modifiers [48]. Many disease proteins aggregate gradually and only trigger pathological changes as the animals age [46]. Thus, on a defined threshold day, forward and reverse genetic screens for enhancers and suppressors of aggregation or toxicity can be performed [49,50,51]. In a similar fashion, *C. elegans* is also suitable for large compound screens, although the impermeability of the animals’ cuticle is a major challenge [52].

In addition to unbiased screens, a number of studies in *C. elegans* are also devoted to elucidating the function of *C. elegans* orthologs of human neurodegenerative disease genes, since the lower redundancy of the nematode genome increases the likelihood for gene knockdowns and knockouts to produce a specific phenotype. Furthermore, work in *C. elegans* has also investigated the influence of genes that have been linked in genome-wide association studies to the pathogenesis of a disease or to a known disease-related protein [53].

Studies in *C. elegans* have clearly contributed to the identification of several modulators and protective substances. Recapitulating the results of these studies in other disease models, including *Drosophila* or mammalian systems, verified that *C. elegans* can be used to gain valuable information about the molecular mechanisms of prion-like diseases and to screen for pharmaceutical compounds that halt or prevent their progression. Several reviews describe in great depth how *C. elegans* has been employed to study the function, aggregation and toxicity of proteins associated with neurodegenerative diseases [54,55,56,57]. Here, we will complement these articles by providing an overview of the literature that specifically models the propagation and intercellular transmission of prions and prion-like proteins in the nematode system.

### 2.2. C. Elegans as a Model System to Study the Prion-Like Propagation of Disease Proteins

According to the concept of self-propagation by seeding, the spreading of protein aggregation involves the transfer of misfolded protein species from the initially affected cell to neighboring cells, where they induce the templated conversion of native proteins. The ability to restrict expression of fluorescently tagged proteins to specific cell types combined with the ease of in vivo imaging, makes the transparent worm particularly useful for tracking proteins in a whole organism in vivo. This led to the recent development of several models that specifically follow how prions and prion-like proteins spread from the expressing donor cells to recipient cells and tissues. Furthermore, a precisely controlled local expression allows the separation of cell autonomous from non-cell autonomous toxicity. 

So far, prion-like spreading of the yeast prion domain of Sup35 (NM), α-Syn, and HTT Exon1-polyQ97 as well as the non-cell autonomous toxicity associated with transthyretin (TTR) amyloidoses have been modeled in *C. elegans* (Table 1). These studies tracked the transmission of the prion-like protein under investigation either by direct observation of fluorescence signal in tissues that do not express the transgene [58,59,60,61] or by the use of bimolecular fluorescence complementation (BiFC) (Figure 1). For the latter case, α-Syn or HTT Exon1-polyQ97 tagged with one non-fluorescent half of a fluorescent protein (enhanced GFP (EGFP) or Venus) were expressed in a subset of neurons and the same proteins tagged with the non-fluorescent complementary other half expressed in the neighboring pharynx [62,63,64] or in an interconnected neuronal population [65]. The appearance of fluorescence signal in the neurons and the pharynx could be attributed to the transmission and subsequent complementation of the fluorescent protein fragments attached to the respective prion-like proteins. One thing to consider with the BiFC models is that they cannot distinguish whether an increase in fluorescence is due to increased transmission or lower turnover of the protein in the donor or recipient cells. Most likely, it is often a combination of both effects, which makes the exact interpretation of the results somewhat difficult. In combination with complementary assays, which are essential for a thorough analysis anyway, they nevertheless offer excellent models for studying the intercellular spreading of disease proteins. 

In the next sections, we will discuss in more detail how these transgenic *C. elegans* models have been employed to gain insights into the mechanisms underlying the prion-like propagation of protein misfolding.

#### 2.2.1. The Autophagy-Lysosomal Pathway as Route for Prion-Like Spreading in *C. Elegans*

An important question in prion-like propagation is which cellular pathways facilitate the release from donor and subsequent uptake of misfolded amyloidogenic proteins into the recipient cells. The first study that investigated this in *C. elegans* made use of the well characterized prion domain NM of the yeast prion protein Sup35 [58]. In addition to the WT domain (NM), two variants were employed, one highly aggregation prone (R2E2) and one not aggregation prone (RΔ2-5) [66], whereby NM and especially R2E2 aggregates were found to be highly toxic. Expression of R2E2 in muscle cells affected the muscle sarcomere and mitochondrial integrity and led to an induction of autophagy. Subsequent co-localization analysis showed that R2E2 primarily localized to vesicles of the autophagy-lysosomal pathway (ALP), namely autophagosomes, late endosomes and in particular lysosomes but not early endosomes, indicating that it was specifically targeted for lysosomal degradation. Likewise, α-Syn is also directed to the ALP and accumulates in the endo-lysosomal system in donor and receiving cells [60]. Strikingly, R2E2 and α-Syn containing vesicles had an unusual elongated shape, indicating that they might undergo frequent fusion, fission and sorting processes and therefore could be different from bona fide lysosomes.

Using these *C. elegans* models, intercellular spreading of R2E2 and α-Syn has been demonstrated from various donor cells and tissues [58,60]. Specifically, transfer has been observed between muscle cells, from muscle cells to hypodermis and coelomocytes (scavenger cells that endocytose material from the pseudocoelom), from the intestine to muscle cells and coelomocytes, and from dopaminergic neurons to the hypodermis, coelomocytes, and occasionally muscle cells [58,60]. Since these proteins were also detected in relatively remote tissues with no direct contact sites, such as hypodermis or coelomocytes in the case of muscle cell-specific expression, it is likely that they follow an exo-endocytosis pathway rather than a direct intercellular transport. Furthermore, several hits (i.e., *apa-2, hgrs-1, vha-15,* and *cct-2*) that inhibited α-Syn spreading in a candidate RNAi screen are linked to vesicular trafficking, endo-lysosomal sorting, maturation, and exo-endocytosis, suggesting an important role of these pathways in the dissemination of misfolded proteins.

**Table 1 biomolecules-10-01188-t001:** *C. elegans* models to investigate prion-like spreading.

Constructs ^1^	Approach to Monitor Spreading	Promotors/Donor Tissue	Receiving/Affected Tissue	Reference
NM::YFP NM RΔ2-5::YFP NM R2E2::YFP/mRFP	detection of fluorescence signal in non-expressing cells by confocal microscopy	myo-3p/BWM vha-6p/intestine	intestine, BWM, coelomocytes	Nussbaum-Krammer et al. [58]
V1::SNCA SNCA::V2::DsRed	BiFC	flp-21p/ADL, ASE, ASH, URA, MC, M2 and M4 neurons myo-2p/pharynx	bidirectional between donor tissues	Kim et al. [64] Bae et al. [62]
EGFP_1–155_::SNCA SNCA::EGFP_156–238_	BiFC	ddr-2p/I1, M3, PVP, VC4, VNC and DNC neurons tph-1p/serotonergic neurons	bidirectional between donor tissues	Tyson et al. [65]
SNCA::YFP SNCA::mRFP SNCA^A53T^::mRFP	detection of fluorescence signal in non-expressing cells by confocal microscopy	myo-3p/BWM dat-1p/dopaminergic neurons	hypodermis	Sandhof et al. [60] Tittelmeier et al. [61]
V1::HTT Ex1-polyQ_25/97_ HTT-Ex1-polyQ_25/97_::V2::DsRed	BiFC	flp-21p/ADL, ASE, ASH, URA, MC, M2 and M4 neurons myo-2p/pharynx	bidirectional between donor tissues	Kim et al. [63]
TTR WT TTR V30M TTR D18G TTR T119M	TTR specific fluorogenic probe; indirect (non-cell autonomous toxicity)	unc-54p/BWM	coelomocytes, FLP neurons	Madhivanan et al. [67]

^1^ NM, prion domain of yeast Sup35; RΔ2–5, NM with deletion of the oligorepeat region; R2E2, NM with extended oligorepeat region; YFP, yellow fluorescent protein; mRFP, monomeric red fluorescent protein; V1/2, N- and C-terminal part of split Venus; SNCA, α-synuclein; BiFC, Bimolecular fluorescence complementation (detection of fluorescence in two specific cell types, each expressing the prion-like protein fused to either of two complementary fluorescence protein fragments by confocal microscopy); Ex1, Exon 1; polyQ, expanded polyglutamine; BWM, body wall muscle; TTR, transthyretin.

Further support for the significance of exo-endocytosis and the ALP in prion-like propagation comes from studies by Kim et al. and Tyson et al., who investigated neuron-pharynx or neuron-neuron transmission of α-Syn in *C. elegans* by BiFC, respectively [64,65]. A decrease in endocytosis (*dyn-1* mutant) reduced α-Syn transmission to the pharynx [64], and modulating synaptic vesicle release impacted neuron-to-neuron transmission [65]. Moreover, inhibition auf autophagy by RNAi or by bafilomycin treatment and impairment of lysosomal degradation capacity by lowering caspase levels (*asp-4* and *asp-1* mutants) increased BiFC fluorescence. Induction of autophagy by rapamycin and the stimulation of lysosomal gene expression either genetically (*hlh-30/TFEB* over expression) or pharmacologically (by N-acetylglucosamine (GlcNAc), treatment) reduced it [64,65]. In addition, RNAi-mediated knockdown of several PD-associated risk genes involved in the ALP (*catp-6*/*ATP13A2*, *lrk-1*/*LRRK family*, and *vps-35*/*VPS35*) increased α-Syn transmission between neurons [65]. Furthermore, *C. elegans* neurons were recently shown to dispose aggregated proteins or damaged organelles by expelling large ‘trash’ filled vesicles (exopheres) during proteotoxic stress, such as aging or impaired autophagy [59]. The extruded material was subsequently found in the hypodermis and coelomocytes [59].

The fact that all these *C. elegans* models independently link autophagic flux and endo-lysosomal capacity of donor and recipient cells to the accumulation and spreading of prion-like proteins underlines the importance of these pathways in this context. A collapse of autophagy and the failure to degrade lysosomal content in donor cells seems to lead to the secretion and eventually transfer of non-degradable material into still healthy neighboring tissues for remote degradation. Accordingly, lysosomal clearance appears to be coordinated between cells within a multicellular organism. The central role of exo-endocytosis and the ALP in the inter-cellular transmission of prion-like proteins in *C. elegans* is consistent with observations made in cell culture, other animal models, and patients. Most prion-like proteins such as α-Syn, Tau, TDP-43, and HTT are intracellular proteins and yet have been shown to enter the endo-lysosomal system, which seems to facilitate their transfer to neighboring cells [68].

The finding that modulating synaptic vesicle release influences transsynaptic spreading of α-Syn between *C. elegans* neurons [65] also reflects the observation in other model systems. For instance, changes in neuronal activity correlate with the release of α-Syn and Tau from mouse primary neurons [69,70] and enhanced neuronal activity exacerbates tauopathy in an AD mouse model [70]. Moreover, exosome-mediated secretion has become a major pathway through which disease proteins are released from cells [71,72,73,74,75,76]. Accordingly, the depletion of the ESCRT-0 component *hgrs-1*, which encodes an ortholog of human hepatocyte growth factor-regulated tyrosine kinase substrate (HGS), impairs α-Syn spreading in *C. elegans* [60], likely due to its role in the formation of multivesicular bodies (MVBs) [77].

Exosomes are derived from intraluminal vesicles (ILVs), which are formed in late endosomes by inward budding of their membrane thereby generating MVBs, a special form of late endosomes. Fusion of an MVB with the plasma membrane subsequently releases ILVs as exosomes into the extracellular space. Exosomal secretion is known to be coordinated with the ALP [78] and therefore it is not surprising that several studies that investigate spreading of disease-related proteins in other model systems have established a robust link between the ALP and the release of these proteins from cells [18,79,80,81,82]. In particular, the exosomal release of α-Syn is enhanced under reduced ALP function [83,84,85]. Inhibition of various steps of the ALP (autophagosome formation [86], autophagosome-lysosome fusion [87], and lysosomal acidification [86]) increased the release of α-Syn in cell culture models, while upregulating autophagic flux decreased it [86]. Similarly, induction of autophagy by rapamycin treatment reduced release of Tau from primary neurons, while inhibition of lysosomal degradation by bafilomycin treatment enhanced it [88]. In contrast, HTT1-571-polyQ72 secretion was impaired by chemical ablation of lysosomes and was dependent on synaptotagmin 7, a regulator of lysosomal secretion, suggesting that this protein is preferentially released by lysosomes rather than exosomes [89]. With respect to TDP-43, cell culture studies have implicated the ALP [90] and an endocytosis dependent endo-lysosomal pathway [91] in the clearance of TDP-43 aggregates but did not investigate a potential effect on TDP-43 secretion into the media. 

After release, the next step in transmission of prion-like proteins is their uptake into receiving cells. Several entry routes have been reported. The mode of uptake likely depends on how the material is released in the first place and might differ between free and exosomal proteins. Exosomes, as means for transcellular cargo delivery have only recently gained scientific interest and their uptake is not fully understood yet. In principle, release of their content could occur by fusion with the plasma membrane of the receiving cell, but more evidence points to non-selective endocytic uptake and entry into the endo-lysosomal pathway [92]. Instead of exosomes, most studies investigating trans-cellular spreading utilized free monomers, oligomers and/or amyloid fibers of various length. Consistent with the observations that genetic inhibition of endocytosis reduces α-Syn transmission in *C. elegans* [60,64], inhibition of endocytosis reduces uptake of α-Syn [93], Tau [94], and HTT Exon1-polyQ44 [95] also in cell culture models. Similarly, neuron to neuron transmission of HTT Exon1-12-polyQ138 in the *Drosophila* brain depends on exo-endocytosis [96].

Finally, the transferred prion-like proteins need to escape from endo-lysosomal vesicles in order to seed the aggregation of naïve proteins in the cytosol of the receiving cell. Apart from TDP-43, direct evidence for the induction of endosomal membrane rupture has been reported for α-Syn, Tau, and HTT Exon1-polyQ45 from several cell culture studies [97,98,99,100]. The same was recapitulated in *C. elegans*, where α-Syn transmitted from dopaminergic neurons or muscle cells efficiently induces endo-lysosomal rupture in the recipient hypodermis [60]. Moreover, this also applies in particular to the BiFC transgenic *C. elegans,* since the fluorescence signal in these transmission models depends on the physical interaction between endogenously expressed and transmitted material.

Taken together, the significance of the ALP for the prion-like spreading of disease proteins in the *C. elegans* models is in agreement with observations in other model systems. Moreover, the data in *C. elegans* also fit to the emerging perception that an overload of the ALP due to chronic accumulation of misfolded proteins, together with a decrease in its capacity due to aging, which is exacerbated by mutations in ALP-related genes, is part of the reason why the progression of several neurodegenerative diseases with prion-like characteristics is accelerated late in life [101,102].

#### 2.2.2. Aging and Insulin Signaling as Modulators of Prion-Like Transmission and Toxicity

Aging is a major risk factor for the development and progression of neurodegenerative diseases. The fact that a strong correlation between increasing age and progressive toxicity and transmission can be observed in the *C. elegans* models underlines their importance for studying prion-like spreading of disease proteins. In the neuron-to-pharynx model of Kim et al., the α-Syn BiFC signal accumulated over the animal’s lifetime and led to the formation of pharyngeal inclusions in older animals [64]. Additionally, the authors observed an age-dependent increase in axonal blebbing and reduction of pharyngeal pumping rates as well as an overall shortened lifespan [64]. In a similar study from the same group that investigated HTT Exon1-polyQ97 spreading, the authors observed the same correlation [63]. α-Syn transmission in the neuron-to-neuron spreading model of Tyson et al. started between day one and five of adulthood, after which the BiFC fluorescence continued to increase [65]. Likewise, expression of mRFP-tagged α-Syn caused cell autonomous toxicity in the expressing muscle cells and dopaminergic neurons, which worsened over time, and resulted in an age-dependent transmission of the protein into the hypodermis [60].

The influence of aging on the spreading of prion-like proteins is further illustrated by the impact of pathways that regulate aging. One of the best-studied pathways that modulate lifespan is the insulin signaling pathway. The important relationship between reduction of insulin signaling and the resulting increase in lifespan was first observed in *C. elegans* [103] and has since been reported in several other models up to mammals [104]. The key receptor in this pathway, DAF-2, is the sole *C. elegans* homologue of the human insulin receptor family and mutating the corresponding gene abolishes insulin signaling and doubles the lifespan of *C. elegans* [105]. Introducing this mutation into the α-Syn::mRFP spreading models, delayed α-Syn spreading and rescued α-Syn related toxicity phenotypes [60]. On the other hand, inhibiting one of DAF-2’s main downstream transcription factors, DAF-16/FOXO [103], accelerates organismal aging and increased α-Syn transmission, aggregation and toxicity [60,64]. In light of the data discussed in the previous section on the importance of a well-functioning ALP for the degradation of pathological protein species and prevention of prion-like propagation, it is not surprising that lifespan extending pathways often enhance the capacity of the ALP. As central signaling pathway monitoring nutrient availability and the nutritional status of the organism, the insulin signaling pathway is directly interlinked with autophagy [106]. Several rodent disease models reveal a protective function of autophagy upregulation following dietary interventions [107] and in a primate PD model, caloric restriction resulted in a reduction of neurochemical deficits and motor dysfunction [108].

An important insight into the beneficial effect of reduced insulin signaling on α-Syn propagation is again provided by studies in *C. elegans*. Interestingly, there seem to be two main protective effects. First, the inhibition of the insulin signaling pathway leads to enhanced clearance of prion-like proteins due to higher autophagic turnover [64] and second, lysosomes become more resilient to the membrane disrupting ability of disease proteins [60]. How exactly these functions are regulated upon blocking the insulin receptor are important open questions that need to be further investigated in the *C. elegans* prion models.

#### 2.2.3. Molecular Chaperones as Cofactors for Prion-Like Propagation 

The identification of potential cofactors involved in the prion-like amplification and intercellular transmission of amyloidogenic proteins is of high relevance as it may lead to new therapeutic approaches. In yeast, several molecular chaperones can affect prion propagation [109,110]. Molecular chaperones are essential components of the cellular protein quality control network, which assist in the correct folding of proteins, prevent their misfolding, and ensure that terminally aggregated proteins are correctly eliminated [111]. In particular, the AAA+ disaggregase Hsp104 is an essential cofactor for prion propagation, since deletion of the corresponding gene results in the rapid loss of yeast prions [112]. Hsp104 collaborates with the Hsp70 system to dissolve aggregated proteins [113,114]. However, by extracting individual monomers, they fragment large prion fibers into smaller seeds, creating more ends for templated incorporation of the soluble isoform. This ultimately promotes the propagation of the prion state rather than preventing it. In line with these findings in yeast, in vitro data and mathematical models also suggest that fragmentation of amyloid fibrils is essential for efficient prion propagation [115,116,117,118].

Since there is no direct homolog of Hsp104 in multicellular organisms, disaggregation function in metazoan cells is provided by the Hsp70 system, whereby an Hsp110-type nucleotide exchange factor (NEF) cooperates with the core HSC70/HSPA8 and a class B J-domain protein [119,120]. This Hsp70 disaggregation machinery is able to dissociate recombinant amyloid fibers composed of α-Syn, Tau, or HTT Exon1-polyQ48 proteins as well as detergent-insoluble protein species extracted from various model systems in vitro [121,122,123], which has raised the question of whether this activity is cytoprotective by dissolving protein aggregates or whether, on the contrary, it contributes to amyloid amplification by generating smaller fragments similar to the role of Hsp104 in yeast prion propagation. *C. elegans* proved to be an excellent model to investigate this question. The nematode chaperone repertoire is less redundant than that of humans, having only one cytosolic Hsp110-type NEF, HSP-110, instead of three in humans, which allowed a specific impairment of the Hsp70 disaggregase by depleting the crucial component HSP-110. Strikingly, the knockdown of HSP-110 significantly decreased the accumulation and toxicity of α-Syn and polyQ35 and reduced the formation of spreading competent α-Syn species [61]. These results imply that the Hsp70 disaggregation machinery is required for the prion-like propagation of disease-associated amyloidogenic proteins in metazoans [61]. The fact that the disaggregation of amyloid-type substrates generated particles that are more toxic and preferred substrates for intercellular spreading suggests that an intervention to reduce their chaperone-mediated remodeling could interfere with their vicious replication cycle. This could be a promising therapeutic strategy to halt disease progression. However, chaperones have several essential cellular functions, which makes them challenging drug targets. Further research is therefore necessary to find means to specifically prevent the disaggregation of amyloids while leaving other activities unaffected.

#### 2.2.4. Cell Autonomous and Non-Cell Autonomous Toxicity of Prion-Like Proteins

The aggregation of disease-associated proteins is usually accompanied by their mislocalization and most likely also by a loss of endogenous function. This might account for some of the cytotoxicity of these proteins. However, the expression of these proteins is toxic in various models, even in the absence of an endogenous homologue [124,125,126], strongly suggesting that a toxic gain-of-function plays a major role in disease pathology. 

Numerous *C. elegans* studies already established the inherent cell autonomous toxic potential of disease-associated proteins [45,127,128,129]. This has been also recapitulated in the more recent models for prion-like propagation. The accumulation of R2E2, for instance, resulted in the disruption of sarcomere structure and mitochondrial integrity in the expressing muscle cells [58]. Mitochondrial integrity is often affected also in other model systems for disease proteins, especially α-Syn, leading to oxidative stress [130,131]. Mitochondrial fragmentation is an active highly regulated reversible process, which is thought to be crucial for the maintenance of a functional mitochondrial network through the removal of damaged parts [132]. Maintaining a well-connected mitochondrial network is thought to be generally beneficial for its energy production [133]. Therefore, excessive fragmentation is particularly harmful to cells with high energy demands such as neurons [134,135]. Accordingly, the inhibition of mitochondrial fragmentation slows neurodegeneration in AD and ALS mouse models [136,137]. These findings exemplify that *C. elegans* is a suitable model to study gain-of-function toxicity of proteins linked to neurodegenerative diseases.

In addition to this expected cell autonomous toxicity, prion-like proteins, however, often exhibit non-cell autonomous toxicity affecting surrounding cells and tissues. This might be partly due to the fact that these proteins are transferred to adjacent cells. Furthermore, the evolution of multicellular organisms has led to the development of cells and tissues with highly specialized functions, with the consequence that impairment of the function of one cell type can also affect the function of interconnected or interdependent cells. This increased complexity also resulted in the emergence of a systemic control of cellular stress responses at the organismal level involving tissues that are distant from the actual site of proteostasis disturbance (extensively reviewed in [138,139]). 

The recently developed nematode models for prion-like propagation are now investigating the toxicity faced by cells and tissues in the vicinity of expressing cells. In the R2E2 prion model, disturbance of cellular structures extended from the expressing muscle cells into the recipient intestine, where mitochondrial fragmentation and a loss of lipid storage droplets was observed [58]. Moreover, the *C. elegans* model for transthyretin (TTR) amyloidoses specifically addresses non-cell autonomous toxicity of extracellular TTR aggregates. In these diseases, the native TTR tetramer, which usually circulates in the blood and central nervous system fluid, is destabilized by particular mutations, and the monomers misfold and form amyloids, which are systemically deposited and mainly damage cardiac tissue and neurons [140]. The expression of disease-related TTR mutants in *C. elegans* muscle cells not only led to cell autonomous toxicity but also affected multiple neurons, resulting in dendritic abnormalities of polymodal nociceptive FLP neurons, and a shortening of mitochondria in mechanosensory neurons [67]. Analogous observations were made in *Drosophila*, where the transmission of pathological HTT Exon1-12-polyQ138 induced non-cell autonomous cell death [96] and widespread apoptosis activation [141] in the brain, although its expression was restricted to olfactory receptor neurons.

The BiFC models make it difficult to differentiate between cell-autonomous and non-cell autonomous toxicity, since in this approach the proteins of interest are expressed in both donor and receiving cells [64,65]. Nevertheless, observations in these model systems suggest that non-cell autonomous mechanisms could also contribute to the reported phenotypes. The expression of α-Syn in the pharynx alone led to a significantly reduced pharyngeal pumping rate as well as to a reduced lifespan compared to WT animals [64]. While the effect on the pumping rate is likely a consequence of cell autonomous toxicity, the reduction in lifespan seems to reflect non-cell autonomous toxicity, since fewer pumping rates are known to reduce food intake, which should rather lead to an extended lifespan [142].

Although there is increasing evidence that α-Syn exerts non-cell autonomous toxicity [143,144], the underlying mechanisms are not yet understood. A decisive advantage of the *C. elegans* models that express the fluorescently tagged disease protein in only one distinct tissue type is the ability to distinguish precisely between its effects on donor and receiving tissue. This way it could be clearly shown that the expression of α-Syn damages endo-lysosomal vesicles not only in the expressing muscle but also in the receiving hypodermis [60]. Although endosomal vesicle rupture is a crucial step in prion-like propagation by allowing seeds to enter the cytosol and template the conversion of its endogenous native counterpart [99] as discussed above, it is also harmful in itself. Lysosomal membrane permeabilization can induce mitochondrial damage and oxidative stress and eventually lead to cell death [145,146]. Hence, the continuous transfer of toxic protein species seems sufficient to damage the receiving epithelial cells without the replication of pathological proteins and the occurrence of amyloid deposits. Interestingly, epithelial dysfunction and blood-brain-barrier leakage has been described for PD and other prion-like disorders [147,148]. Even if α-Syn was present in the receiving cells, it is conceivable that the constant influx of lysosome damaging material could challenge the cellular proteostasis network to such an extent that it can no longer counteract the aggregation of the cell endogenous α-Syn. This may cause α-Syn misfolding in the receiving cells without the need for templated seeding. 

In recent years, the purely neuron centric view of the etiology in neurodegenerative diseases has shifted and the involvement of non-neuronal glial cells has gained significant scientific ground. They are now not only recognized as victims, but also as a source of non-cell autonomous toxicity. Microglia, the CNS residing macrophages, were shown to be involved in the spreading of several prion-like proteins [149,150,151,152,153]. Moreover, the uptake of toxic protein species might facilitate microglia activation and lead to neuroinflammation, which is implicated in several neurodegenerative diseases [154]. Indeed, microglial activation and synapse loss preceded the appearance of neurofibrillary tangles (intracellular large amyloid deposits of Tau) in a tauopathy mouse model [155], indicating that either stress signals from the affected neurons or the transmission of smaller Tau oligomers into microglia initiates an inflammatory response [156]. Similarly, oligomeric α-Syn can be endocytosed by astrocytes, the most abundant type of glia in the brain, which results in inclusion formation, activation of microglia, and inflammation [157,158].

Gaining mechanistic insights into these intricate relationships between neurons and non-neuronal cells is crucial for understanding neurodegenerative diseases. However, it is difficult to systemically characterize and manipulate distinct cell types in complex animal models. Through the precise cell type specific expression of disease related proteins and reporters of stress response pathways, as well as through well-established cell type- and tissue-specific phenotypic assays, *C. elegans* offers the opportunity to further investigate how neighboring cells and tissues react to local protein misfolding and how stress responses and the clearance of aggregated proteins are coordinated across tissue boundaries. Its reduced genetic complexity also eases the analysis of how organism wide signaling pathways, such as the *daf-2* pathway, modulate not only cell autonomous but also non-cell autonomous toxicity.

## 3. Future Perspectives and Concluding Remarks

The novel *C. elegans* models discussed here show great potential for the identification and delineation of pathways that influence the propagation and spreading of prions and prion-like proteins (Figure 2). On the one hand, several pathways that affect prion-like propagation in other model systems had conserved effects in *C. elegans*, validating its usefulness. On the other hand, the fact that some important aspects may not be sufficiently investigated in other model systems, such as the influence of the ageing process or the impact of the extracellular matrix, further argues for the application of the nematode system as a complementary model system. For example, a recent study identified 57 novel modulators of extracellular proteostasis in *C. elegans* whose knock down increased protein aggregation in the intercellular space [159]. A potential impact of these extracellular proteostasis regulators on the propagation and non-cell autonomous toxicity of prion-like proteins would be most efficiently studied in the *C. elegans* prion models. In particular, proteinopathies involving extracellular protein misfolding, such as TTR amyloidoses and AD, could benefit from these studies, as they could reveal novel treatment options.

An important technical challenge that prevents the use of these models in genome-wide genetic screens and large drug screens is the microscopy-based readout of changes in fluorescence intensities in a small area of the animal over its lifespan. Currently this requires the repeated mounting of animals onto microscope slides for high-resolution imaging, which is not feasible on a large scale. However, this could be overcome by implementing the recent advances in microfluidics for high throughput imaging and phenotypic testing of *C. elegans* [57,160]. Long-term analysis will also allow to distinguish modifiers that only delay spreading from those that inhibit transmission throughout the whole lifetime.

In addition, the restriction of RNAi to specific cell types for tissue- and cell type-specific knockdown of candidates [161] will make it possible to distinguish modifiers that act in donor, recipient or both tissues. Similarly, recent advances in cell type specific enrichment methods for tissue specific transcriptome analysis [162,163,164] will help to elucidate whether cells react differently to the expression compared to the influx of toxic prion-like proteins. The same methods could also be used to decipher cell autonomous, non-cell autonomous, and systemic responses of the proteostasis network to local protein misfolding stress.

Single cell analysis together with the toolbox of well-defined cell type specific promoters might finally also help us to understand the mechanisms underlying the selective vulnerability of specific neurons to different prion-like proteins. It would be interesting to extend the BiFC approach to different populations of interconnected neurons and analyze whether neuron-to-neuron transmission is a random equally distributed process or whether it depends on the recipient neuron type. Likewise, following the fate of the recipient neurons fate throughout the animal’s lifespan might reveal if neurodegeneration purely correlates with the amount of transmitted material or also with the neuron type.

In the face of an ever-growing number of people affected by neurodegenerative diseases and the almost complete lack of pharmaceutical treatment options, the need for easily screenable models for prion-like propagation increases. *C. elegans* systems could pave the way for the discovery of new targets and drugs for therapeutic intervention. Fast and affordable screening for genetic modifiers or potential drugs that affect prion-like propagation in an intact metazoan organism will help to narrow down promising candidates.

Like any model system, *C. elegans* has its inherent weaknesses that limit its usefulness for the investigation of neurodegenerative diseases. The nematode lacks several internal organs and tissues, and its neural network is far from having the complexity of a real brain. Therefore, both drug targets and compounds discovered or tested in *C. elegans* need to be validated for their translatability, safety, and efficiency in mammalian cell culture and animal models to progress towards the ultimate goal of developing treatments that prevent or halt neurodegeneration.

## Figures and Tables

**Figure 1 biomolecules-10-01188-f001:**
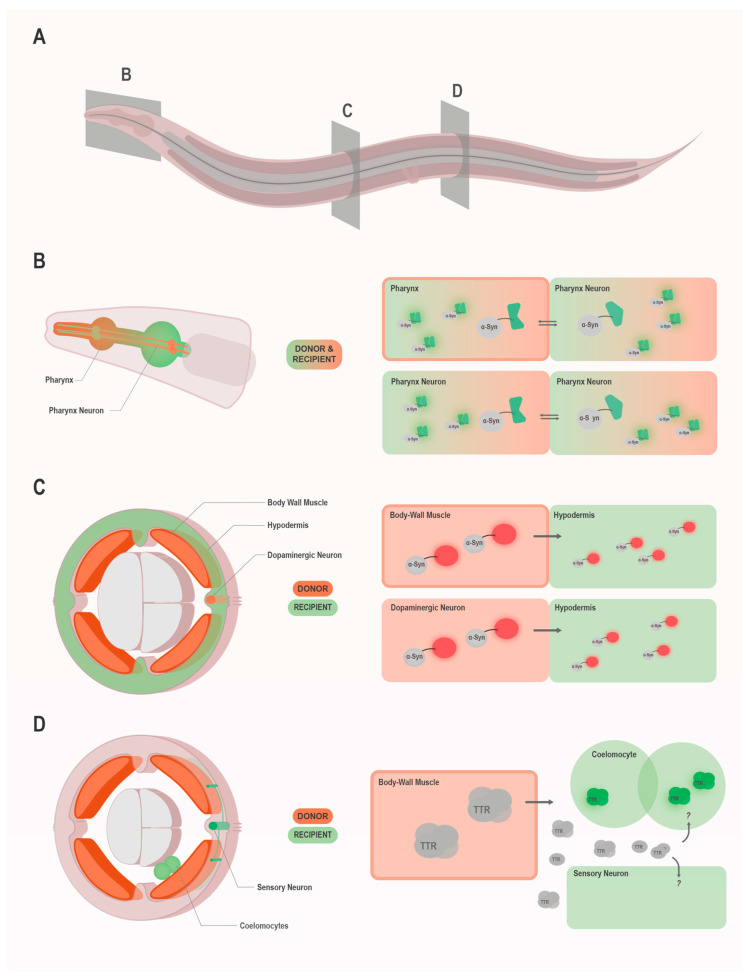
Overview of the currently established *C. elegans* models for prion-like propagation. The three different types of model systems and their working principles are illustrated using α-Syn and TTR as examples. Further models for other prion-like proteins are described in the text and listed in Table 1. (**A**) Schematic overview of *C. elegans* with the locations of the longitudinal and cross-sections, which are shown in more detail in panel B–D. (**B**–**D**) Longitudinal (**B**) or cross-sections (**C**,**D**) on the left show exemplary cells of the donor & recipient tissues within the worm. On the right-side, expression and transmission of the respective prion-like protein is shown schematically. Thick edges, as seen in pharynx and body wall muscle indicate presence of a tissue border (basement membrane). Large symbols represent construct that is expressed in the respective cell, while smaller symbols represent transmitted proteins and proteins formed after transmission. (**B**) Representation of *C. elegans* models using the bimolecular fluorescence complementation (BiFC) method to study the spreading of α-Syn. Here, α-Syn expressed in either the pharynx or pharyngeal neurons (or different neuronal subsets) is coupled with separate fluorophore fragments (e.g., Venus or GFP), which only exhibit fluorescence upon excitation, when they complement to a functional protein. Close proximity of the fragments, as in α-Syn aggregation, is required for complementation to happen. Thus, fluorescence observed in either cell indicates intercellular exchange of the α-Syn, as well as close proximity. (**C**) Representation of *C. elegans* models using fluorescent proteins (here mRFP) fused to α-Syn to track transmission from the donor tissue, in which expression takes place (here the body wall muscle or dopaminergic neurons), to surrounding recipient tissues (here hypodermis). (**D**) Representation of *C. elegans* models in which transthyretin (TTR) is expressed without any tag in the donor tissue (here the body wall muscle). Transmission into recipient tissues, such as the coelomocytes, is then studied by the addition of a compound which binds TTR tetramers. This binding can then be visualized by fluorescence microscopy. Non-cell autonomous effects of TTR secretion and accumulation in or near sensory neurons, can be indirectly inferred by behavioral assays reporting on sensory neuron function.

**Figure 2 biomolecules-10-01188-f002:**
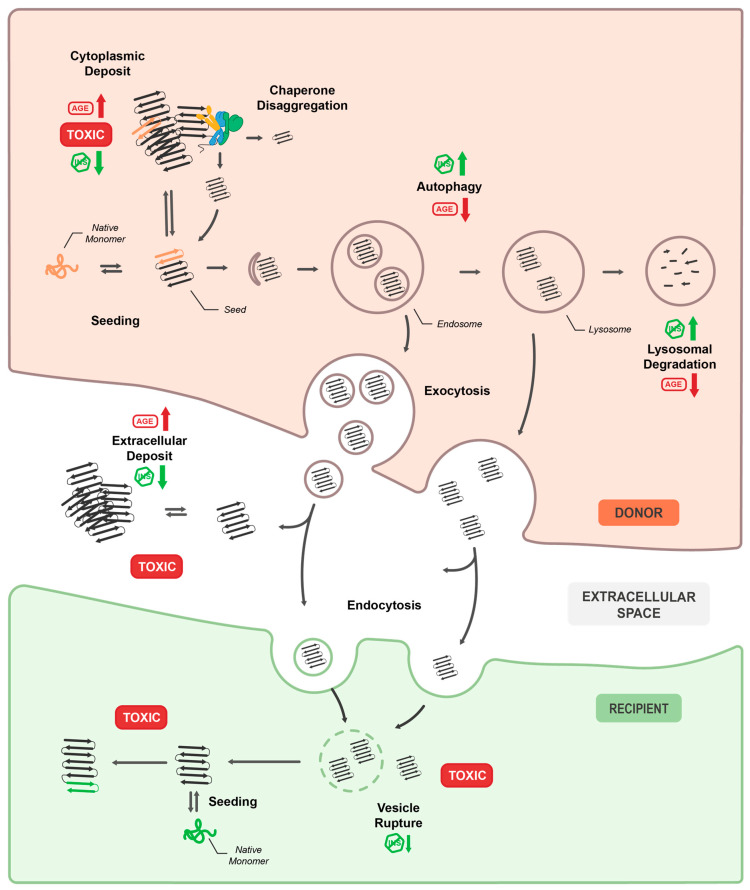
Schematic overview of the cellular pathways involved in the propagation of prion-like aggregates within and between *C. elegans* tissues. In the donor tissue, native monomers may spontaneously or due to cellular stress acquire a pathological β-sheet rich conformation, which in turn can seed the misfolding of cognate monomers and thus grow into larger fibrils that eventually deposit into large cytoplasmic inclusions. While the exact nature of the toxic protein species is still unclear, the process of aggregate formation has been clearly linked to cell autonomous toxicity. The cellular chaperone network attempts to dissociate protein aggregates, resulting in the generation of additional seeds that may promote prion-like propagation. Misfolded and aggregated protein species are recognized by the autophagy machinery and subsequently either targeted for lysosomal degradation or removal via exocytosis. Aggregated material expelled from the donor tissue can end up in extracellular deposits that can damage adjacent tissues non-cell autonomously, or it can be taken up via endocytosis into the surrounding tissue. From the endocytic compartment of the recipient cell, seeds may enter the cytosol after vesicle rupture. Once in the recipient cytoplasm, misfolding of cognate native monomers can be templated by the seeds, which also confers non-cell autonomous toxicity. Factors known to impact prion-like propagation and toxicity, are for example ageing (AGE) and the impairment of insulin signaling (INS), the effects of which are indicated in the figure at relevant locations.

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
