# Peer review of "C. elegans Models to Study the Propagation of Prions and Prion-Like Proteins"

_biomolecules, 2020, doi:10.3390/biom10081188_

Round 1

Reviewer 1 Report

This is a nicely-written and timely review thoroughly summarizing current literature using C. elegans models to study prion-like protein transmission, a topic of growing interest especially concerning age-related neurodegeneration. The authors discuss advantages of using nematode models to study basic mechanisms underlying prion-like diseases and provide great detail about genes and pathways that have been implicated in these studies. The authors discuss many proposed mechanisms for how pathological proteins could spread between different cells based on studies in C. elegans and other models; however, an additional figure illustrating these potential mechanisms (e.g., lysosomal escape, endo/exocytic transport, relevance to aging pathways, proteostasis, and non-cell autonomous toxicity) would be extremely helpful.

A second concern is that though the authors touch briefly on this, more discussion of the relevance and possible limitations of using an invertebrate model that lacks a true brain to study diseases that largely affect the central nervous system would be appropriate.

Minor comments:

  • Avoid use of the word “prove” (used 3 times in text)
  • In Fig. 1B, the coloring in the cartoon suggests that protein expression in donor and recipient cells is continuous. How could inter-cellular transmission be assessed if this is the case?
  • In Fig. 1, use of size to indicate proteins originating in donor cells vs transmitted is not clear. For example, in Fig. 1C, the smaller apparently transmitted species appear in both donor and recipient cells, yet the arrows suggest unidirectional movement.

Additional comments:

There are several references that should be refer in the authors’ discussion of roles for glia in prion-like spreading:

1. Donnelly KM et al (2020) doi: 10.7554/eLife.58499

2. George S et al (2019) doi: 10.1186/s13024-019-0335-3

3. Hopp SC et al (2018) doi: 10.1186/s12974-018-1309-z

4. Pearce MMP et al (2015) doi: 10.1038/ncomms7768

Reviewer 2 Report

In this review Sandhof and collegues well summarised how the use of C. elegans can help in elucidate the mechanisms underlying the propagation of diseases related to protein misfolding. The manuscript is well organised, clearly written and the properly considers the relevant openquestions in the field of prion-like proteins transmission. I have no major considerations but only a suggestion. Recent data has provide evidence that experimental brain trauma induces a self-propagating tau pathology, which can be transmitted between mice in a prion-like manner (Zanier ER et al., Brain, 2018). In the paragraph 1.1 of the Introduction, TBI should be considered as a tau prion related neurodegenerative patholgy. 
